# Recent Advances in the Diagnosis of Classical Swine Fever and Future Perspectives

**DOI:** 10.3390/pathogens9080658

**Published:** 2020-08-15

**Authors:** Lihua Wang, Rachel Madera, Yuzhen Li, David Scott McVey, Barbara S. Drolet, Jishu Shi

**Affiliations:** 1Department of Anatomy and Physiology, College of Veterinary Medicine, Kansas State University, Manhattan, KS 66506, USA; rachelmadera@vet.k-state.edu (R.M.); yuzhen@vet.k-state.edu (Y.L.); 2United States Department of Agriculture, Arthropod-Borne Animal Diseases Research Unit, Center for Grain and Animal Health Research, Manhattan, KS 66502, USA; scott.mcvey@usda.gov (D.S.M.); barbara.drolet@usda.gov (B.S.D.)

**Keywords:** classical swine fever, laboratory diagnosis, technologies, future perspectives

## Abstract

Classical swine fever (CSF) is a highly contagious viral disease of pigs, including wild boar. It is regarded as one of the major problems in the pig industry as it is still endemic in many regions of the world and has the potential to cause devastating epidemics, particularly in countries free of the disease. Rapid and reliable diagnosis is of utmost importance in the control of CSF. Since clinical presentations of CSF are highly variable and may be confused with other viral diseases in pigs, laboratory diagnosis is indispensable for an unambiguous diagnosis. On an international level, well-established diagnostic tests of CSF such as virus isolation, fluorescent antibody test (FAT), antigen capture antibody enzyme-linked immunosorbent assay (ELISA), reverse-transcription polymerase chain reaction (RT-PCR), virus neutralization test (VNT), and antibody ELISA have been described in detail in the OIE Terrestrial Manual. However, improved CSF diagnostic methods or alternatives based on modern technologies have been developed in recent years. This review thus presents recent advances in the diagnosis of CSF and future perspectives.

## 1. Introduction

Classical swine fever (CSF), a list-A disease classified by the World Organization for Animal Health (OIE), is considered as a transboundary animal disease by the Food and Agriculture Organization of the United Nations (FAO) [1]. The disease causes high morbidity and mortality in both feral and domestic pigs and can result in significant economic losses to the swine industry worldwide [2]. Currently, it is present in many countries in Asia, the Caribbean islands, Africa, and South and Central America (Figure 1). It is most likely to be introduced to CSF-free countries through inadvertent or deliberate importation of classical swine fever virus (CSFV) infected animals, animal products, and animal feed [2,3].

Classical swine fever virus (CSFV) is the etiologic agent of CSF and belongs to the genus *Pestivirus* in the *Flaviviridae* family [4]. The genome of CSFV is a positive single-strand RNA of about 12.3 kb. It contains untranslated regions at 5′ and 3′ ends and a single large open reading frame (ORF). The ORF codes four structural (C, E^rns^, E1, and E2) and eight nonstructural viral proteins (N^pro^, p7, NS2, NS3, NS4A, NS4B, NS5A, and NS5B) [5,6]. Based on the nucleotide sequences of 5′-non-translated region (5′-NTR) and glycoprotein E2, CSFVs are divided into three genotypes and 11 sub-genotypes (1.1–1.4, 2.1–2.3, and 3.1–3.4) [7,8,9]. As reported, CSFV genotype 2.1 and genotype 2.3 caused the more recent outbreaks in Europe [10]. Sub-genotypes 1.1, 2.1, 2.2, and 2.3 are prevalent in Asia [11], while sub-genotypes 3.1-3.4 are distributed in other separated geographic regions [1,12,13,14].

Traditional diagnostics for CSF include clinical signs, pathological findings, and antigen and antibody detection [15]. Although unique clinical and pathological observations such as “button” ulcers in the cecum and large intestine mucosa may be found exclusively in CSF, other clinical signs and pathological findings in pigs infected with CSFV are highly variable and are often similar to that of other viral diseases of pigs, such as African swine fever, pseudorabies, porcine dermatitis, and nephropathy syndrome (PDNS), post-weaning multisystemic wasting syndrome (PMWS), thrombocytopenic purpura, and various septicemic conditions [16]. Thus, laboratory diagnosis of CSF for detection of the specific CSFV antigen and antibody is indispensable [15,16]. The well-established diagnostic methods of CSF such as virus isolation, fluorescent antibody test (FAT), antigen capture antibody enzyme-linked immunosorbent assay (ELISA), reverse-transcription polymerase chain reaction (RT-PCR), virus neutralization test (VNT), and antibody ELISA (Table 1) have been widely used and well described in the OIE Terrestrial Manual [17]. Recently developed techniques and alternatives have made significant improvements in several key components of CSF diagnosis, including less sample and reagents required, less effort and time needed, increased detection efficiency (multiplexing), ease of performing and disposal, automation, and point of care (POC). This review provides an updated overview on laboratory diagnosis of CSF and future perspectives.

## 2. Antigen Detection

### 2.1. Virus Isolation

Virus isolation in cell culture is the oldest laboratory technique for detecting CSFV. Porcine kidney cell lines (PK-15 and SK-6) are often used for isolation of CSFV [17]. However, the use of other porcine cells including swine primary cells (pulmonary alveolar macrophages and peripheral blood mononuclear cells) may enhance the chances of obtaining different CSFVs with different growth characteristics. Since CSFV does not cause a cytopathic effect (CPE), the growth of CSFV in the cells is usually visualized by using immunological technologies with fluorescent or horseradish peroxidase (HRF)-conjugated antibodies [17,18,19].

The cell culture, virus propagation, and staining are labor intensive and time-consuming (weeks). In addition, skilled and experienced personnel and adequate facilities are needed for cell culture, handling CSFVs and accurate interpretation of the CPE. These disadvantages make virus isolation less attractive for mass surveillance or rapid diagnosis. However, virus isolation is still considered the “gold standard” for confirming CSF clinical cases and the only method for making virus collections (Table 1).

### 2.2. Fluorescence Antibody Test (FAT)

FAT is the commonly used staining method for CSFV detection. It utilizes fluorescein isothiocyanate (FITC) labeled antibodies to detect CSFV proteins in the slices of cryostat (frozen) tissues or fixed cells. Anti-CSFV gamma-globulins prepared from specific pathogen-free pigs are recommended to be used. These globulins can ensure that most variant CSFVs will be captured. The differentiation of CSFV from other *pestiviruses* in FAT positive samples, especially bovine viral diarrhea virus (BVDV) and border disease virus (BDV), can be done using RT-PCR with genetic typing or virus isolation in cell culture with specific monoclonal antibody (mAb) typing [17,19].

The main advantages of FAT are that it is relatively easy and rapid to perform and allows direct visualization of the CSFVs in stained tissues. Therefore, it is useful for a first laboratory investigation in suspected clinical cases (Table 1). Several FITC conjugated anti-CSFV antibodies (polyclonal or monoclonal) for FAT are commercially available for research purposes, such as those from Creative Diagnostics, Bioss Inc., Biorbyt LLC, and so on. However, FAT requires highly specialized equipment (i.e., fluorescent microscope) and immunohistochemical staining expertise. It is only recommended to be used in laboratories that have the expertise of performing this technique. The novel ViewRNA in situ hybridization method can detect CSFV RNA directly in infected cells [20]. Using RNA in an in situ hybridization method and specific probes of CSFV RNA, the relative location of CSFV RNA can be visualized in PK15 cells. The sensitivity of this method was three to four orders of magnitude higher than that of FAT. The specificity experiment showed that it was highly specific for CSFV (sub-genotypes 1.1, 2.1, 2.2, and 2.3) and without cross-reaction with other *pestiviruses* including BVDV, porcine parvovirus (PPV), porcine pseudorabies virus (PRV), and porcine circovirus II (PCV-2). This assay has the potential to be used for testing for CSFV in cells. However, it remains to be determined whether this method can be used to detect CSFV in swine tissues and it is still expensive and is not commercially available yet.

### 2.3. Antigen-Capture ELISA

Antigen-capture ELISA uses anti-CSFV antibodies on an ELISA plate to capture the CSFV proteins [21]. It has been developed for the rapid screening of large numbers of pigs with clinical suspicion of CSFV infection [15,17,21,22,23]. Commercial antigen-capture ELISA kits are available from several commercial vendors including IDEXX Laboratories, Thermo Fisher Scientific, MEDIAN Diagnostics, and so on. These kits are double-antibody-sandwich (DAS)-based ELISA for detecting CSFV E2 or E^rns^ protein in serum, blood, plasma, or tissue extracts (Table 2). 

Antigen-capture ELISA is fast (provides results within 4 h), easy to perform, and does not require specialized equipment. It can be applied at a herd level for confirmation of clinical cases or determining infection-free population status (Table 1). However, its sensitivity and specificity are lower than most of the other diagnostics, especially the real-time RT-PCR. It is not recommended for testing individual animals and has been increasingly discouraged in recent years.

### 2.4. Real-Time Reverse Transcription Polymerase Chain Reaction (Real-Time RT-PCR)

Real-time RT-PCR has now replaced the traditional RT-PCR and has become an essential tool in the routine diagnosis of CSFV [24,25,26]. It is a suitable approach for confirmation of clinical cases and prevalence of infection surveillance for CSF (Table 1). Several commercial real-time RT-PCR kits are available for rapid and specific detection of CSFV RNA, including IDEXX RealPCR CSFV RNA Mix, virotype^®^ CSFV RT-PCR Kit, CSFV dtec-RT-qPCR Test, ADIAVET™ CSF REAL TIME, CSFV genesig^®^ Advanced and standard kits, and so on (Table 3). These kits use either SYBR green or TaqMan probe to detect the accumulation of amplicon during the exponential phase of the reaction, which can specifically and sensitively test the CSFV in serum, blood, plasma, viral culture, tissue, or swabs.

The disadvantages of real-time RT-PCR are its high cost and complexity due to simultaneous thermal cycling and fluorescence detection, false positives caused by laboratory contamination from polluted specimens or equipment, and false negatives caused by PCR inhibitors in the sample or degraded RNA [27]. The improved real-time RT-PCRs and advanced alternatives have been designed to help resolve these issues. One-step and automated RT-PCRs can reduce the risk of contamination [27,28,29]. The primer-probe energy transfer RT-PCR assay provides a higher specificity by analyzing the melting curve following PCR amplification [30,31]. The loop-mediated isothermal amplification (LAMP) assay can accumulate the CSFV amplicon under isothermal conditions [32,33,34]. The functionalized gold nanoparticles were developed as nanoflare probes for rapid detection of CSFV without nucleic acid amplification [35].

Multiplex real-time RT-PCR as a powerful technique has expanded exponentially in the diagnosis of CSF in recent years. It is quite useful and convenient for quick and accurate detection of different pathogens in mixed infections, which is common in swine production systems. Multiplex RT-PCR assays for rapid detection and genotyping of CSFVs [36,37], simultaneous detection, and differentiation of common swine viruses [38,39,40,41] have been developed. Additionally, multiplex combined high-throughput molecular diagnostic platform, user-friendly electronic microarray, magnetoelastic sensor, and microfluidic detection systems were developed as potential alternatives for detection and surveillance of CSFV infection [42,43,44,45,46]. These assays can save considerable time and effort without compromising robustness and sensitivity and can reduce the sample and reagent requirement as well. 

### 2.5. Next Generation Sequencing (NGS)

Next Generation Sequencing (NGS) is a highly sensitive method for generating sequence data and exploring the genetic characters of infectious agents. It has been extensively applied to metagenomics and whole-genome sequencing of infectious viral diseases of livestock [47,48]. By using NGS, researchers found that there might be a long-term persistence of genotype 2.3 CSFV strains in wild boar in Germany [49]. By analyzing NGS data of CSFV isolates of varying virulence in infected pigs, higher quasispecies diversity and more nucleotide variability were found in viral samples from pigs infected with the highly virulent isolates compared to samples of pigs infected with low and moderately virulent isolates [50]. Evolutionary changes in virus populations following the challenge of naïve and vaccinated pigs with the highly virulent CSFV strain were studied using the NGS technology and this study found that vaccination imposes a strong selective pressure on CSF viruses that subsequently replicate within the vaccinated animals [51].

The complete genome sequences obtained from NGS can provide detailed genetic information for construction of reliable phylogenetic relationships of CSFVs for monitoring the evolution and transmission patterns during field outbreaks or epidemics of CSF. One phylogenetic analysis using 58 CSFV complete genome sequences from different Asian countries indicated that the circulating Indian CSFV strains belong to different branches of the 1.1 sub-genotype [52]. These data combined those obtained from other different diagnostic tests can be used for meta-analysis of CSF prevalence, which is important for the investigation of CSF prevalence in different regions [52].

Currently, most of the NGS platforms are expensive to establish and require highly skilled molecular biologists and bioinformaticians. The implementation of NGS is still a challenge and cannot be used as a routine test for disease diagnosis due to cost and the time required [47,48]. However, with the novel and emerging sequencing technologies, cost-effective, user-friendly, and portable NGS will be developed and will act as an effective tool for CSF control and prevention.

## 3. Antibody Detection

### 3.1. Virus Neutralization Test (VNT)

VNT is the gold standard for sensitivity and specificity of antibody detection methods. It can be used for confirmation of clinical cases, prevalence of infection surveillance, evaluation of the immune status post-vaccination, and the efficacy of CSF vaccines (Table 1) [53,54,55]. However, VNT is a work-intensive and time-consuming procedure that requires cell culture and a high-containment laboratory that can handle infectious CSF virus. In addition, it cannot be automated, thus it is not suitable for mass analysis of samples [14,26,53,54,55].

More recently, alternatives have been developed to overcome the disadvantages of VNT. A neutralizing mAb-based competitive ELISA (cELISA) with emphasis on the replacement of VNT for C-strain post–vaccination monitoring was developed in our group. The test principle of this cELISA is that the neutralizing mAb can compete with C-strain vaccine induced neutralizing antibodies in pig serum to bind the capture antigen C-strain E2 protein. The established cELISA showed 100% sensitivity (95% confidence interval: 94.87 to 100%) and 100% specificity (95% confidence interval: 100 to 100%) when testing C-strain VNT negative pig sera (*n* = 445) and C-strain VNT positive pig sera (*n* = 70) and showed excellent agreement (Kappa = 0.957) with VNT when testing the pig sera (*n* = 139) in parallel. The inhibition rate of serum samples in the cELISA is highly correlated with their titers in VNT (r^2^ = 0.903, *p* < 0.001). The C-strain antibody can be tested in pigs as early as 7 days post vaccination with the cELISA. This cELISA is a reliable, rapid, simple, safe, and cost-effective tool for sero-monitoring of C-strain vaccination at a population level [56]. In addition, another group developed a high-throughput VNT by using the recombinant CSFV possessing a small report tag and luciferase system. As reported, the VNT titers of the serum can be determined tentatively at 2 days post-infection (dpi) and are comparable to those obtained by conventional VNTs at 3 or 4 dpi. This system allows CSF virus growth to be easily and rapidly monitored and enabled the rapid and easy determination of the VNT titer using a luminometer, which could be a powerful tool to replace the conventional VNT as a high-throughput antibody test for CSFV infections [57].

### 3.2. Antibody ELISA

Antibody ELISA is the quickest, easiest, and most widely used technique for serological diagnosis and epidemiological investigation of CSF. It is suitable for herd or individual animal CSFV infection screening, prevalence of infection surveillance, and immune status checking in individual animals or populations post-vaccination (Table 1). The E2 protein is crucial for inducing an immune response in the host following CSFV infection [58]. Detection of E2 antibodies in the serum of animals is an easy and reliable method for monitoring CSFV infection during and after outbreaks and for testing coverage of immunization after vaccination [59,60,61].

Several commercial CSF antibody ELISA kits are available including those from Biocheck, Boehringer Ingelheim, Cusabio Technology LLC, IDEXX Laboratories, ID VET, Indical Bioscience, iNtRON Biotechnology, Median Diagnostics Inc., Thermo Fisher Scientific, and so on. Most of the commercial kits are indirect, competitive, or blocking ELISAs based on the detection of envelop glycoprotein E2 specific antibodies (Table 4). Limitations of these antibody ELISAs are lower specificity (i.e., cross-reactions with BVDV, BDV, and other *pestiviruses*) and inability to discriminate animals vaccinated with conventional attenuated vaccines or E2-based subunit vaccines.

## 4. Differentiation of Infected from Vaccinated Animals (DIVA) Diagnostic Methods

### 4.1. Genetic DIVA

The rationale of genetic DIVA (differentiation of infected from vaccinated animals) is the identification of genetic differences between vaccine strains and wild-type CSFVs. Both traditional RT-PCR and real-time RT-PCR (single-plex or multiplex)-based CSF genetic DIVA systems have been developed and evaluated [62,63,64,65,66,67,68,69]. Multiplex nested RT-PCR and real-time RT-PCR assays have been developed for differential detection of wild-type virus from C-strain vaccine [62,63,64,65,66,67,68]. A one-step RT-PCR using TaqMan minor-groove-binding (MGB) probes was developed to distinguish between attenuated Korean LOM and wild-type strains of CSFV in Korea [69]. A simple RT-PCR based on the T-rich insertions in CSFV genome was developed for rapid differentiation of wild-type and at least three attenuated lapinized vaccine strains [70]. The modified genotype 1.1 (including C-strain) real-time RT-PCR assay with a real-time RT-PCR assay that detects all known CSFV strains has been successfully used to distinguish C-strain vaccine from the circulating field strains that do not belong to genotype 1 [71].

The genetic DIVA approach facilitates a rapid and reliable differentiation of field virus infected from live attenuated virus vaccinated domestic pigs and wild boars. It is especially useful for detection of the infected animals that are incompletely protected by vaccination and will play a critical role for making decisions prior to and during cessation of a control strategy that employs vaccination with CSF live vaccines.

### 4.2. Serological DIVA

The ideal serological DIVA test has the ability to discriminate antibodies induced by CSFV infection from the vaccine-derived antibodies, so it can rule out CSFV infected pigs from vaccinated pigs [72]. This can be obtained by detection of specific antibodies against antigens or epitopes that are modified or lacking in a subunit or marker vaccine. It has been shown that antibodies to E^rns^ can be used as an indicator of CSFV infection in pigs and the E^rns^-based ELISA can be used as a companion diagnostic test to identify CSFV-infected pigs vaccinated with the E2-based subunit or marker vaccines [73,74,75,76]. Currently, two E^rns^ ELISAs are commercially available and have been evaluated as accompanying DIVA diagnostic tools for E2 subunit vaccines, CP7_E2alf, or similar chimeric vaccines. One is prioCHECK CSFV E^rns^ (Thermo Fisher Scientific, Waltham, MA, USA); the other is pigtype CSFV E^rns^ Ab (Indical Bioscience, GMBH, Leipzig, Germany) (Table 4). Published data on their evaluations showed that prioCHECK CSFV E^rns^ has a sensitivity of 90–98% with sera from CSFV infected domestic pigs and a specificity of 89–96% with sera from vaccinated domestic pigs [77]. In combination with the marker vaccine “CP7_E2alf”, pigtype CSFV E^rns^ Ab has a sensitivity of 90.2% and a specificity of 93.8% [78]. However, cross-reactivity with antibodies against other *pestiviruses* was observed for these two E^rns^ ELISAs [77,78]. Depending on the represented data, these two E^rns^ ELISAs are recommended to be used on a herd basis and not for diagnostic analysis on samples of single animals.

Additional approaches or alternatives are undergoing development or further optimization. These include the multiplex microsphere immunoassay [79], which is capable of discrimination within epitope-specific antibody populations [80] and the indirect E^rns^ antibody ELISA with *Pichia pastoris-*expressed E^rns^ [81]. Recently, our research group successfully generated a mAb against E^rns^, which can specifically recognize C-strain, but not react with wild-type CSFVs or other viruses in the genus *Pestivirus*. A cELISA was developed in our group based on the strategy that the C-strain-specific mAb will compete with the C-strain vaccine-induced antibodies in pig serum to bind the capture antigen (C-strain E^rns^) [unpublished data]. Different from the CSFV neutralizing monoclonal anti-E2 antibody based cELISA for sero-monitoring of C-strain vaccination at a population level [56], this novel anti-E^rns^ mAb-based cELISA is a valuable tool for measuring and differentiating immune responses to C-strain vaccination and/or infection in pigs. The data about the establishment and validation of this C-strain specific cELISA will be published separately at a later date. In brief, suitable tools for serological DIVA are available. However, there is room for improvement, especially with respect to cross-reactivity issues.

## 5. Point-of-Care (POC) Diagnostics

User-friendly, cost-effective, rapid, and reliable POC diagnostics (i.e., diagnosis of diseases directly on-site) are indispensable tools for immediate decisions of effective and evidence-based disease control strategies [82]. For example, dipstick tests are designed to use thin paper/plastic strips coated with specific antiviral antibodies to detect viral antigens in serum and other body fluids. Lateral flow assays (LFA) and microfluidic devices are two different and yet more complex technologies that are also based on the biochemical interaction of antigen–antibody. The principles for these three immunochromatographic assays are the same as sandwich ELISA and the major difference between them is that the immunological reaction is carried out on different platforms for different assays. POC studies in animal health management are rare compared to human and companion animal medicine. The immunochromatographic assay-based kits including Antigen Rapid Test Kit (Ring Biotechnology Co., Ltd., Beijing, China), LiliF™ CSFV antibody rapid test kit (iNtRON Biotechnology, Inc., Gyeonggi, South Korea), and CSFV Antibodies Rapid Test Kit (Antibodies-online Inc., Limerick, PA, USA) are commercially available for rapid testing of CSFV antigen or antibodies in the field. Laboratory-based assays, including the loop-mediated isothermal amplification, combined with a lateral flow dipstick assay [83], the immunochromatographic strip [84], and duplex lateral flow assay [85] have been investigated as potential CSF POC tools as well.

POC diagnostics showed advantages in rapidity and portability, which are the most important parameters considered by farmers and veterinarians [86]. It is foreseeable that as interests and needs of stakeholders increase and new portable POC technologies emerge, novel and applicable POC diagnostics will be developed for detection, control, and prevention of CSF in the field in the near future.

## 6. Future Perspectives

Although commercial and in-house diagnostics (antigen detection and antibody detection) of CSF are available, there is still room for improvement. The authors suggest that the following aspects should be considered: (i) Continuously improving the sensitivity, specificity, costs, speed, automation, and POC is necessary; (ii) Reference materials (serum bank, virus bank, and non-infectious molecular standards) should be produced and be accessible for validation of the developed CSF diagnostics; (iii) The development of DIVA diagnostics without cross-reaction with antibodies induced by other *pestiviruses* is critical.

The other emerging infectious diseases, such as the spreading of African swine fever in Asia and the ongoing pandemic of Coronavirus disease 2019 (COVID-19), may shift focus away from the CSF [87,88]. However, as long as CSF exists, it will remain a continuous threat to the pig industry worldwide. Therefore, international cooperation on surveillance and control of CSF becomes even more crucial, both currently and in the future. Researchers should continue to work on developing novel rapid and reliable diagnostics to facilitate the surveillance and control of CSF.

## Figures and Tables

**Figure 1 pathogens-09-00658-f001:**
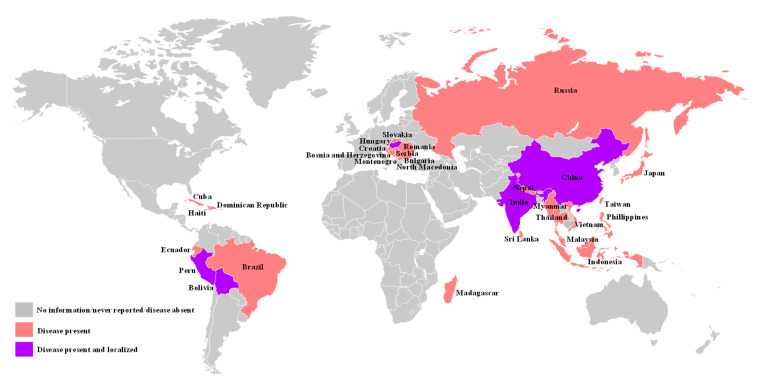
Global distribution of classical swine fever (CSF) epidemics, 2020. Map based on data from CABI Invasive Species Compendium. Wallingford, UK: CAB International. Available online: www.cabi.org/isc (accessed on 05292020). In addition, we also incorporated the most current CSF epidemic information (disease present in Japan and Romania) from OIE, 2020. https://www.oie.int/animal-health-in-the-world/official-disease-status/classical-swine-fever/map-of-csf-official-status/ Names of countries with CSF are given in the map.

**Table 1 pathogens-09-00658-t001:** Well-established CSF diagnostic methods and their application.

Method	Application	Advantages	Disadvantages
Virus Isolation	Confirmation of clinical cases; Making virus collections; May be used for individual animal freedom from infection prior to movement	“reference standard"; Very sensitive; Indicates active infection	Work intensive and time consuming; Requires specialized microscope and expertise
FAT ^1^	Confirmation of clinical cases	Quick and direct visualization of antigens in tissue	Requires specialized equipment, expertise, and confirmatory test
Antigen-capture ELISA ^2^	Population infection-free status; May be used for confirmation of clinical cases	Fast, does not require specialized equipment and suitable for herd screening	Low sensitivity; Cross-reactivity with other Pestiviruses
RT-PCR ^3^	Confirmation of clinical cases; Prevalence of infection surveillance; May be used for population or individual animal freedom from infection prior to movement	Fast, sensitive, and specific	Specialized equipment; Possibility for false negative results due to sample degradation
VNT ^4^	Individual animal infection-free status prior to movement; Prevalence of infection-surveillance; Immune status in individual animals or populations post-vaccination; Confirmation of clinical cases; May be used for population freedom from infection	Gold standard for sensitivity and specificity	Work intensive and time consuming; Requires specialized microscope and expertise
Antibody ELISA	Population freedom from infection; Individual animal freedom from infection prior to movement; Prevalence of infection-surveillance; Immune status in individual animals or populations post-vaccination	Fast, does not require specialized equipment and suitable for herd screening	Cross-reactivity with other Pestiviruses

Note: Table adapted from Table 1 in 2019 OIE Terrestrial Manual [17]; ^1^ Fluorescent antibody test; ^2^ Enzyme-linked immunosorbent assay; ^3^ Reverse-transcription polymerase chain reaction; ^4^ Virus neutralization test.

**Table 2 pathogens-09-00658-t002:** List of representatives of commercially available CSF antigen-capture ELISA kits/reagents.

Name	Producer	Test Principle	Suitable Sample Materials	DIVA Potential	Web Site
IDEXX CSFV Ag Serum Plus	IDEXX Laboratories, Inc.	DAS ELISA test E^rns^	Serum, plasma, tissue	Yes	https://www.idexx.com/en/livestock/livestock-tests/swine-tests/idexx-csfv-ag-serum-plus-test/
PrioCHECK™ CSFV Antigen ELISA kit	Thermo Fisher Scientific, Inc.	DAS ELISA test E2	Serum, blood, plasma, leukocyte concentrate, tissue extract	No	https://www.thermofisher.com/order/catalog/product/7610047?SID=srch-srp-7610047#/7610047?SID=srch-srp-7610047
VDPro® CSFV AG ELISA	MEDIAN Diagnostics Inc.	DAS ELISA test E2	Cell cultures, leukocyte concentrate, tissue extract	No	http://www.mediandiagnostics.com/eng/es-csf-02.php

Note: DAS, Double-antibody-sandwich; ELISA, enzyme-linked immunosorbent assay; DIVA, differentiate infected from vaccinated animals.

**Table 3 pathogens-09-00658-t003:** List of representatives of commercially available CSF Real-time RT-PCR kits/reagents.

Name	Producer	Test Principle	Suitable Sample Materials	Web Site
ADIAVET™ CSF REAL TIME	BioMérieux	Real-time RT-PCR test CSFV RNA	Serum, blood, viral culture, tissue	https://www.biomerieux-nordic.com/csfv-classical-swine-fever
CSFV dtec-RT- qPCR Test	Genetic PCR solutions™	Real-time RT-PCR test CSFV RNA	Serum, plasma, blood, viral culture, tissue	http://www.geneticpcr.com/index.php/en/pathogen-r-d-qpcr/classical-swine-fever-virus
CSFV genesig® Kits	Primerdesign^TM^ Ltd	Real-time RT-PCR test CSFV RNA	Serum, plasma, blood, viral culture, tissue	https://www.genesig.com/products/9770-classical-swine-fever-virus
CSFV Real Time RT-PCR Kit	Creative Biogene	Real-time RT-PCR test CSFV RNA	Serum, plasma, tissue	https://www.creative-biogene.com/Classical-Swine-Fever-Virus-CSFV-Real-Time-RT-PCR-Kit-PDAS-AR002-1290596-88.html
Classical swine fever virus detection kits	Bioingentech Ltd	Real-time RT-PCR test CSFV RNA	Serum, blood, viral culture, tissue	https://www.kitpcr.com/pcr-kit/classical-swine-fever-virus-detection-kits/
IDEXX RealPCR CSFV RNA Mix	IDEXX Laboratories, Inc.	Real-time RT-PCR test CSFV RNA	Serum, plasma, blood, viral culture, tissue	https://www.idexx.com/en/livestock/livestock-tests/swine-tests/realpcr-csfv/
virotype® CSFV RT-PCR Kit	Indical Bioscience, GMBH	Real-time RT-PCR test CSFV RNA	Serum, plasma, blood, viral culture, tissue	https://www.indical.com/products/assays/

Note: RT-PCR, reverse-transcription polymerase chain reaction.

**Table 4 pathogens-09-00658-t004:** List of representatives of commercially available CSF antibody ELISA kits/reagents.

Name	Producer	Test Principle	Suitable Sample Materials	DIVA Potential	Web Site
BioChek CSFV E2 Antibody ELISA	Biocheck	Indirect ELISA test E2 antibodies	Serum	No	https://www.biochek.com/swine-elisa/classical-swine-fever-antibody-test-kit/
Classical Swine Fever Virus Antibody(IgG) ELISA Kit	Cusabio Technology LLC	Indirect ELISA test CSFV antibodies	Serum	No	https://www.cusabio.com/ELISA-Kit/Classical-Swine-Fever-Virus-AntibodyIgG--ELISA-Kit-114911.html
IDEXX CSFV Ab	IDEXX Laboratories, Inc.	Blocking ELISA test CSFV antibodies	Serum, plasma	No	https://www.idexx.com/en/livestock/livestock-tests/swine-tests/idexx-csfv-ab-test/
ID Screen©Classical Swine Fever E2 Competition	ID VET	Competitive ELISA test E2 antibodies	Serum, plasma	No	https://www.id-vet.com/produit/id-screen-classical-swine-fever-e2-competition/
LiliF™ Classical Swine Fever virus Ab rapid test kit	iNtRON Biotechnology, Inc.	Lateral flow immuno-chromatographic assay test CSFV antibodies	Blood	No	https://intronbio.com:6001/intronbioen/product/product_view.php?PRDT_ID=1891&page=1&Scate1=2&Scate2=2&Scate3=4&Scate4=16&Scate5=1&Scate6=-91-&Sword=
Pigtype CSFV Erns ELISA	Indical Bioscience, GMBH	Double-antigen ELISA test Erns antibodies	Serum, plasma	Yes	https://www.indical.com/products/assays/
PrioCHECK™ Porcine CSFV Ab 2.0 strip kit	Thermo Fisher Scientific, Inc.	Blocking ELISA (E2 antibodies)	Serum, plasma	No	https://www.thermofisher.com/order/catalog/product/7610600?SID=srch-srp-7610600#/7610600?SID=srch-srp-7610600
PrioCHECK™ CSFV Antibody ELISA kit	Thermo Fisher Scientific, Inc.	Blocking ELISA test E2 antibodies	Serum, plasma	No	https://www.thermofisher.com/order/catalog/product/7610046#/7610046
PrioCHECK™ CSFV Erns Antibody ELISA Kit	Thermo Fisher Scientific, Inc.	Blocking ELISA test Erns antibodies	Serum	Yes	https://www.thermofisher.com/order/catalog/product/7610370#/7610370
SVANOVIR®CSFV-Ab	Boehringer Ingelheim	Indirect ELISA test E2 antibodies	Serum	No	https://www.svanova.com/products/porcine/pp031.html
VDPro® CSFV AB C-ELISA	Median Diagnostics Inc.	Blocking ELISA test E2 antibodies	Serum	No	http://www.mediandiagnostics.com/eng/es-csf-02.php
VDPro® CSFV Erns Ab b-ELISA	Median Diagnostics Inc.	Blocking ELISA test Erns antibodies	Serum	Yes	http://www.mediandiagnostics.com/eng/es-csf-02.php
Classical Swine Fever Virus Antibodies Rapid Test Kit	Antibodies-online Inc	Sandwich GICA test CSFV antibodies	Blood, serum	No	https://www.antibodies-online.com/kit/5708730/Classical+Swine+Fever+Virus+Antibodies+Rapid+Test+Kit/

Note: GICA, Gold Immunochromatography Assay; ELISA, enzyme-linked immunosorbent assay.

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
