# Peer review of "Recent Advances in the Diagnosis of Classical Swine Fever and Future Perspectives"

_pathogens, 2020, doi:10.3390/pathogens9080658_

Round 1

Reviewer 1 Report

This manuscript summarizes the diagnostic tests for classical swine fever (CSF), especially the improved methods or alternatives based on modern technologies. It is interesting and contributes to the development of diagnostic methods for CSF. However, some of the tests and related kits are not well described. Here are some aspects that this reviewer concerns:

Major concerns:

  1. The authors provide some representatives of commercially available kits for CSF, for example, the antibody ELISA. It is better if the authors can show more research data about these kits, especially the comparison among them.
  2. Line 264-270, did authors talk about the reference 56? If not, what is the difference between them?
  3. Line 276-277, provide more details about these dipstick tests, lateral flow assays and microfluidic devices, like the definition or mechanism of them. Also, clarify the connection between these three POC devices and “the immunochromatographic assay-based kits”.

Minor concerns:

  1. Line 30, please provide full name of OIE, not just abbreviation
  2. Line 31, please provide full name of FAO---Food and Agriculture Organization of the United Nations
  3. Line 48, 1.1, 2.1 and 2.3 are sub-genotypes, not genotypes.
  4. Line 49, change “and genotype 3” to “while sub-genotypes 3.1-3.4 are”
  5. Line 98, detail the specialized equipment and expertise.
  6. Line 99-104, detail the mechanism of ViewRNA in situ hybridization; Can this method used for testing the CSFV in the tissue samples?
  7. Line 116, add DIVA to the note under the table. Because DIVA is the first time appearing here
  8. In table 2, PrioCHECK™ CSFV Antigen ELISA kit detects E2 antigen
  9. Line 123, become—becomes
  10. Line 171, only 53 or 58 CSFV complete genome sequences were used in the reference 52. The authors of the reference said 53 in the text, but 58 sequences displayed in the figure. Please double check the number.
  11. Line 258-259, list the references of 78 and 79 right behind of the data of each kit.

Author Response

Dear Reviewer: 

In response to your comments, the following changes have been made to the manuscript. The comments are listed below and followed by a description of how we responded to the comments. All changes in the text are highlighted by the “Track Changes” function and the exact line numbers are also provided below.

Major concerns:

1. “The authors provide some representatives of commercially available kits for CSF, for example, the antibody ELISA. It is better if the authors can show more research data about these kits, especially the comparison among them.”

We appreciate the comments from the reviewer. Table 4 (Lines 225-226) has listed 13 different commercial antibody ELISA kits with companions on the testing principle, suitable sample materials, and DIVA potential. The limitations of these ELISA kits are also discussed in the paragraph (Lines 222-224) immediately above Table 4.

2. “Line 264-270, did authors talk about the reference 56? If not, what is the difference between them?”

We thank the reviewer for this comment. No, the current Lines 271-275 described another ELISA that is different from the one reported in reference 56. This is unpublished data from authors’ group. We have added the following information in Lines 271-275. “Different from the CSFV neutralizing monoclonal anti-E2 antibody based cELISA for sero-monitoring of C-strain vaccination at a population level [56], this novel anti-Erns mAb-based cELISA is a valuable tool for measuring and differentiating immune responses to C-strain vaccination and/or infection in pigs.”

3. Line 276-277, provide more details about these dipstick tests, lateral flow assays and microfluidic devices, like the definition or mechanism of them. Also, clarify the connection between these three POC devices and “the immunochromatographic assay-based kits”.

We appreciate the comment and have added the following information on Lines 283-289. “For example, dipstick tests are designed to use thin paper/plastic strips coated with specific antiviral antibodies to detect viral antigens in serum and other body fluids. Lateral flow assays (LFA) and microfluidic devices are two different and yet more complex technologies that are also based on the biochemical interaction of antigen-antibody. The principles for these three immunochromatographic assays are the same as sandwich ELISA, the major difference is in that the immunological reaction is carried out on different platforms for different assays.”

Minor concerns:

4. Line 30, please provide full name of OIE, not just abbreviation

On Lines 29-30, we have revised the sentence as “a list-A disease classified by the World Organization for Animal Health (OIE)”

5. Line 31, please provide full name of FAO---Food and Agriculture Organization of the United Nations

On Line 31, we have added “of the United Nations” to the sentence.

6. Line 48, 1.1, 2.1 and 2.3 are sub-genotypes, not genotypes.

On Line 49, we have replaced “Genotype” with “Sub-genotypes”.

7. Line 49, change “and genotype 3” to “while sub-genotypes 3.1-3.4 are”

On Line 50, we have replaced “and genotype 3” with “while sub-genotypes 3.1-3.4 are”.

8. Line 98, detail the specialized equipment and expertise.

On Line 101, we have added “(i.e. fluorescent microscope” and “immnohistochemial staining” to define the specialized equipment and expertise.

9. Line 99-104, detail the mechanism of ViewRNA in situ hybridization; Can this method used for testing the CSFV in the tissue samples?

On Line 103-105, we have added the following info to detail the mechanism of ViewRNA: “Using RNA in situ hybridization method and specific probes of CSFVRNA, the relative location of VSFV RNA can be visualized in PK15 cells.”

On Lines 109-110, we have added the following info regarding its usefulness for tissue samples: “it remains to be determined whether this method can be used to detect CSFV in swine tissues, and”

10. Line 116, add DIVA to the note under the table. Because DIVA is the first time appearing here

We have added the definition of DIVA on Lines 121-122.

11. In table 2, PrioCHECK™ CSFV Antigen ELISA kit detects E2 antigen

We have updated Table 2 (Lines 119-120) to show that PrioCHECK™ CSFV Antigen ELISA kit detects E2 antigen.

12. Line 123, become—becomes

On Line 129, we have changes “become” to “becomes”.

13. Line 171, only 53 or 58 CSFV complete genome sequences were used in the reference 52. The authors of the reference said 53 in the text, but 58 sequences displayed in the figure. Please double check the number.

We double checked Figure 1 in reference 52 and confirmed that 58 complete genomes were used in the analysis. Thus, we replaced “67” with “58” on Line 175.

14. Line 258-259, list the references of 78 and 79 right behind of the data of each kit.

On Lines 258-259, we have listed the references of 78 and 79 right behind of the data of each kit (Lines 261 and 262).

In summary, significant changes have been made in the revised manuscript to address your suggestions and concerns. 

Thanks you very much for your thoughtful and constructive reviews.

Sincerely,

Jishu Shi, DVM, PhD

Professor of Vaccine Immunology

Reviewer 2 Report

In this excellent manuscript, the Authors concisely and accessibly describe diagnostic tools used in classical swine fever (CSF) diagnostics. Manuscript is well organised. The authors give an overall good background and review of available methods for diagnostic purposes: beginning from the classic virology to advanced molecular methods (i.e. real-time RT-PCR).

I would suggest to enrich the introduction and specify the clinical and pathological findings of CSF. Some of them like “button” ulcers in the cecum and large intestine mucosa may be found exclusively in CSF.

We must remember that every diagnosis, before samples reach diagnostic laboratories, begins at the level of classic veterinarian surveillance.

Second minor issue is presence of CSF in Japan which is not indicated in the map (Fig.1.) – please consider the map from OIE (https://www.oie.int/animal-health-in-the-world/official-disease-status/classical-swine-fever/map-of-csf-official-status/) Japan still has a suspended status of CSF-free country. CSF appeared in Japan in 2018 (1)

  1. Postel A, Nishi T, Kameyama KI, et al. Reemergence of Classical Swine Fever, Japan, 2018. Emerg Infect Dis. 2019;25(6):1228-1231. doi:10.3201/eid2506.181578

Author Response

Dear Reviewer: 

In response to your comments, the following changes have been made to the manuscript. The comments are listed below and followed by a description of how we responded to the comments. All changes in the text are highlighted by the “Track Changes” function and the exact line numbers are also provided below.

1. In this excellent manuscript, the Authors concisely and accessibly describe diagnostic tools used in classical swine fever (CSF) diagnostics. Manuscript is well organized. The authors give an overall good background and review of available methods for diagnostic purposes: beginning from the classic virology to advanced molecular methods (i.e. real-time RT-PCR).

We thank the reviewer for this comment.

2. I would suggest to enrich the introduction and specify the clinical and pathological findings of CSF. Some of them like “button” ulcers in the cecum and large intestine mucosa may be found exclusively in CSF. We must remember that every diagnosis, before samples reach diagnostic laboratories, begins at the level of classic veterinarian surveillance.

We appreciate this comment from the reviewer and have added the following information to the introduction section on Lines 53-54: “Although unique clinical and pathological observations such as “button” ulcers in the cecum and large intestine mucosa may be found exclusively in CSF, other clinical …”

3. Second minor issue is presence of CSF in Japan which is not indicated in the map (Fig.1.) – please consider the map from OIE (https://www.oie.int/animal-health-in-the-world/official-disease-status/classical-swine-fever/map-of-csf-official-status/) Japan still has a suspended status of CSF-free country. CSF appeared in Japan in 2018 (1). Postel A, Nishi T, Kameyama KI, et al. Reemergence of Classical Swine Fever, Japan, 2018. Emerg Infect Dis. 2019;25(6):1228-1231. doi:10.3201/eid2506.181578

We appreciate this comment from the reviewer and have modified Figure 1 to include Japan and a web link to OIE’s CSF status information (Lines 37-40).

In summary, significant changes have been made in the revised manuscript to address your suggestions and concerns. 

Thanks you very much for your thoughtful and constructive reviews.

Sincerely,

Jishu Shi, DVM, PhD

Professor of Vaccine Immunology

Round 2

Reviewer 1 Report

This revised version presents information well.  Here are two minor comments:

  1. Line 103, change CSFVRNA to CSFV RNA, VSFV to CSFV
  2. Line 285, change "the major difference" to "and the major difference between them"

Author Response

Dear Reviewer: 

In response to your comments, the following changes have been made to the manuscript. The comments are listed below and followed by a description of how we responded to the comments. All changes in the text are highlighted by the “Track Changes” function and the exact line numbers are also provided below.

This revised version presents information well.

We appreciate for this comment.

Minor comments:

  1. Line 103, change CSFVRNA to CSFV RNA, VSFV to CSFV

        On Line 103, we have changed “CSFVRNA” to “CSFV RNA” and changed “VSFV” to “CSFV”.

  1. Line 285, change "the major difference" to "and the major difference between them"

        On Line 285 and Line 286, we have changed "the major difference" to "and the major difference between them".

Thank you very much for all of your thoughtful and constructive reviews.

Sincerely,

Jishu Shi, DVM, PhD

Professor of Vaccine Immunology